# CaBagE: A Cas9-based Background Elimination strategy for targeted, long-read DNA sequencing

Amelia D. Wallace[1,2], Thomas A. Sasani[3], Jordan Swanier[1], Brooke L. Gates[4], Jeff Greenland[4], Brent S. Pedersen[1,2], Katherine E. Varley[4], Aaron R. Quinlan[1,2,5]*

1 Department of Human Genetics, School of Medicine, University of Utah, Salt Lake City, Utah, United States of America, 2 Utah Center for Genetic Discovery, School of Medicine, University of Utah, Salt Lake City, Utah, United States of America, 3 Department of Genome Sciences, University of Washington, Seattle, Washington, United States of America, 4 Department of Oncological Sciences, Huntsman Cancer Institute, Salt Lake City, Utah, United States of America, 5 Department of Biomedical Informatics, School of Medicine, University of Utah, Salt Lake City, Utah, United States of America

* aquinlan@genetics.utah.edu

**Data Availability Statement:** We are not able to make the whole-genome data available for the patient samples from the Coriell Institute, per their guidelines. We can, however, submit the data to

## Abstract

A substantial fraction of the human genome is difficult to interrogate with short-read DNA sequencing technologies due to paralogy, complex haplotype structures, or tandem repeats. Long-read sequencing technologies, such as Oxford Nanopore's MinION, enable direct measurement of complex loci without introducing many of the biases inherent to short-read methods, though they suffer from relatively lower throughput. This limitation has motivated recent efforts to develop amplification-free strategies to target and enrich loci of interest for subsequent sequencing with long reads. Here, we present CaBagE, a method for target enrichment that is efficient and useful for sequencing large, structurally complex targets. The CaBagE method leverages the stable binding of Cas9 to its DNA target to protect desired fragments from digestion with exonuclease. Enriched DNA fragments are then sequenced with Oxford Nanopore's MinION long-read sequencing technology. Enrichment with CaBagE resulted in a median of 116X coverage (range 39–416) of target loci when tested on five genomic targets ranging from 4-20kb in length using healthy donor DNA. Four cancer gene targets were enriched in a single reaction and multiplexed on a single MinION flow cell. We further demonstrate the utility of CaBagE in two ALS patients with *C9orf72* short tandem repeat expansions to produce genotype estimates commensurate with genotypes derived from repeat-primed PCR for each individual. With CaBagE there is a physical enrichment of on-target DNA in a given sample prior to sequencing. This feature allows adaptability across sequencing platforms and potential use as an enrichment strategy for applications beyond sequencing. CaBagE is a rapid enrichment method that can illuminate regions of the 'hidden genome' underlying human disease.

SRA and dbGaP for future access. All sequencing data from heathy donors are available from the Sequence Read Archive (PRJNA687491). Sequencing data generated from NIGMS Human Genetic Cell Repository Samples will be shared through dbGAP because donors did not consent to public posting of personally identifying genetic information. Individual-level data are available for download by authorized investigators: https://view.ncbi.nlm.nih.gov/dbgap-controlled Data dictionaries and variable summaries are available on the dbGaP FTP site: https://ftp.ncbi.nlm.nih.gov/dbgap/studies/phs002368/phs002368.v1.p1/ The public summary-level phenotype data may be browsed at the dbGaP study report page: https://www.ncbi.nlm.nih.gov/projects/gap/cgi-bin/study.cgi?study_id=phs002368.v1.p1 Please refer to the release notes for more details: https://ftp.ncbi.nlm.nih.gov/dbgap/studies/phs002368/phs002368.v1.p1/release_notes/Release_Notes.phs002368.CaBagE_Cas9.v1.p1.MULTI.pdf.

**Funding:** A.D.W. was awarded the NIH Ruth L. Kirschstein National Research Service Award (NRSA)Institutional Training Grant (T32): National Human Genome Research Institute Training in Genomic Medicine to support this work (5T32HG008962-05, PI: Lynn Jorde, https://www.genome.gov). A.R.Q. received the University of Utah Equipment Grant A.R.Q. received the National Institutes of Health R01HG006693 award from the National Human Genome Research Institute (https://www.genome.gov) A.R.Q. received the National Institutes of Health R01GM124355 award from the National Institute of General Medical Sciences (https://www.nigms.nih.gov) The funders had no role in study design, data collection and analysis, decision to publish, or preparation of the manuscript.

**Competing interests:** The authors have declared that no competing interests exist.

## Introduction

While short-read DNA sequencing technologies have enabled the discovery of genetic variants underlying numerous rare genetic disorders [1, 2], a large fraction of the human genome remains very difficult to interrogate with short-reads. These so-called "hidden" regions are difficult to sequence with short technologies owing to a mixture of sequence paralogy, complex haplotype structures, and tandem repeats [3, 4]. Collectively these hidden regions impact over 700 genes [4].

Paralogous sequences consist of ancestrally duplicated genomic segments. These sequences can be entire genes or segmental duplications (a duplicated sequence >1kb) and can appear in tandem or interspersed throughout the genome. Due to high homology elsewhere in the genome, there is ambiguity when mapping short reads to these regions. Thus, approximately 70% of segmental duplications are not sequence-resolved in the human reference genome, and are simply annotated as gaps [5]. Polymorphic mobile element insertions are similarly difficult to map, as multiple copies exist throughout the genome and yet broad phenotypic effects of this variation have been suggested [6, 7].

Short tandem repeats (STRs) are another class of genomic sequence that is difficult to resolve, and have estimated mutation rates orders of magnitude higher than single nucleotide variation [8]. Yet the contribution of tandem repeats to phenotypic heterogeneity remains poorly understood due to limitations in our ability to accurately detect and genotype these features. STR expansions underlie over 40 developmental and neurological disorders [9], highlighting a clear need for better molecular and informatics techniques to genotype these features across individuals [10]. The (CCCCGG)n repeat expansion in *C9orf72* segregates with up to 40% of familial amyotrophic lateral sclerosis (ALS) cases [11] and is one of few established causes of the disease [12]. However, sequencing through complete *C9orf72* repeat expansions is difficult; therefore, diagnostics rely on laborious, semi-quantitative methods such as Southern blot or repeat-primed PCR (RP-PCR). In contrast, long-read sequencing (LRS) can, in principle, provide essential quantitative information such as repeat length and sequence content, which may reveal connections between allelic polymorphism and clinical phenotypes such as severity and age of onset.

Oxford Nanopore Technologies (ONT) long-read sequencing (LRS) [13] enables direct measurement of loci containing complex structures without introducing biases due to amplification or polymerase slippage, and permits highly accurate mapping. At the same time, native modifications to DNA or RNA are preserved and can be detected concurrently with the nucleic acid sequence. While higher error rates limit the accuracy of single nucleotide variant discovery compared to Illumina DNA sequencing, long reads that completely span hidden genomic regions offer the potential for comprehensive and accurate discovery of the structural variation therein. A recent study sequenced fifteen human genomes with long reads and showed that over 80% of structural variants genotyped were missed when called from Illumina data for the same subjects [14]. In fact, the sensitivity of LRS can greatly exceed standard next generation sequencing (NGS), particularly for large insertions (>50bp) [15].

The ONT MinION is particularly advantageous for diagnostics, as it is affordable, portable, and capable of generating reads up to 1Mb. A pressing limitation of the MinION however, is the low throughput relative to other sequencing technologies (e.g., Illumina). This has motivated recent efforts to enrich loci of interest for subsequent LRS without amplification, which limits target-lengths and can introduce PCR bias. Many emerging methods leverage the highly specific targeting ability of the CRISPR/Cas9 system, but strategies vary widely and have unique strengths and limitations related to DNA input requirements, protocol execution time, target size restrictions, and efficiency [16–22]. CATCH was one of the first methods published and relies on pulsed-field gel electrophoresis to physically isolate a DNA target of known size

that is first cut at the flanks with Cas9 [17, 23]. This method is amenable to very large targets (200kb) because DNA is protected from shearing in agarose plugs. However, if the target length is variable or unknown, as with pathogenic repeat expansions, the method suffers and amplification is often required to obtain high sequencing yields. Subsequent strategies improved yield and efficiency by enriching sequencing data for target sequences without physical enrichment of target DNA fragments in the sample. The nCATS method uses dephosphorylation to prevent adapter ligation in sample DNA [24]. Next, the 5-prime phosphates flanking a target are restored using the endonuclease activity of Cas9, so that those fragments alone are available for sequence adapter ligation. This method performs best for targets up to 20-30kb. Most recently, ReadFish, a computational method for real-time enrichment during sequencing, has been expanded to human genomic targets [25]. The method utilizes real-time sequence identification to allow off-target DNA fragments to be rejected from nanopores prior to completion of sequencing, thus performing targeted sequencing without specialized library preparation. ReadFish does not have cost associated with assay design, reagents, or equipment, however rejection of fragments from pores does decrease overall output from flow cells and thus reduces yield across individual targets [25]. Here we introduce a Cas9-based Background Elimination strategy, CaBagE. In contrast to nCATs and ReadFish, CaBagE physically enriches genomic DNA for specific target loci, producing enrichment with comparable efficiency in terms of library preparation time and sequence output. A similar strategy called Negative Enrichment has been independently proposed [26], but with enrichment 3 to 32-fold lower after LRS than with CaBagE.

Cas9 is a single-turnover enzyme with endonuclease activity that can be easily directed to specific genomic sequences using guide RNAs. The complex formed between the enzyme, its RNA guide, and target DNA is very stable, and forcibly dissociates only under harsh environmental conditions [27]. *In vitro* studies have shown that the natural dissociation time of Cas9 from its DNA target is approximately 6 hours [28]. When challenged with competing proteins, Cas9 remains tightly bound in most cases [29]. We were therefore motivated to ask whether this property of Cas9 extends to multiple progressing exonucleases. If so, one can leverage exonucleases as a means to deplete background DNA and enrich for targeted loci that are bound and therefore protected by Cas9 on either side.

Exonucleases have previously been used to eliminate background DNA in NGS libraries [26, 30, 31]. For example, Nested Patch PCR protects target DNA from digestion by capping the target sequences with adapters containing phosphodiester bonds [30] and ChIP-exo protocols rely on proteins bound to DNA to protect the "footprint" from exonuclease activity [31]. By directing Cas9 binding to either side of a specific target locus, we show that the DNA flanked by Cas9 is preserved amidst extensive digestion of genomic DNA by exonucleases, allowing for highly specific target enrichment without PCR. By coupling Cas9-based background elimination with long-read sequencing technology, we demonstrate target sequence enrichment in previously poorly characterized regions of the human genome. Further, we combine this output with a computational approach that allows clustering of long-read sequence alignments to yield genotypes across a pathogenic repeat expansion in *C9orf72*. This generalizable molecular framework is fast, accurate, and multiplex-ready, to characterize recalcitrant yet medically important genes.

## Results

### Cas9 Background Elimination (CaBagE) targeted sequencing strategy overview

To enrich for a genomic region of interest, we developed a method that uses Cas9 to selectively protect target DNA from background elimination by exonucleases (**Fig 1**). First, Cas9 is targeted to both sides of a region of interest using locus-specific guide RNAs. The distance

between the enzymes, effectively the target fragment length, is highly flexible and limited only by the ability to design guide RNAs flanking the target and the average fragment length of source genomic DNA. Immediately following Cas9 binding, Exonucleases I, III, and Lambda are introduced to degrade single stranded DNA, and double-stranded DNA from the 3-prime and 5-prime direction, respectively. These enzymes degrade most DNA present in the sample with the exception of the fragments flanked by the Cas9 enzymes, namely, the DNA target of interest. Heat incubation is then used to inactivate the exonucleases and force dissociation of the Cas9 enzyme from the target DNA. Then, the ends of the target DNA fragments are available for A-tailing and ligation of the sequencing adapters. Sequencing libraries are prepared beginning with the adapter ligation step of the ONT Cas-mediated PCR-free enrichment protocol (developed for use with nCATs) and sequenced on a single MinION flow cell for 48 hours. Target enrichment and library preparation can be completed in approximately 6 hours.

## Cas9 prevents processive exonuclease from degrading DNA target

To test whether bound Cas9 prevents DNA degradation by a combination of three processive exonucleases, a 997bp synthetic double-stranded DNA gBlock (IDT) was designed to contain multiple guide RNA target sites. Cas9 cleavage requires that the target DNA, which is complimentary to the RNA guide, contains a 3bp protospacer adjacent motif (PAM) at its 3′ end. Cas9 binding affinity differs between the PAM-proximal and distal sides of the cleavage site [28]. Therefore, the gBlock was designed such that flanking pairs of target sites could be in either "PAM-in" or "PAM-out" orientation, where the PAM sequences contained in the paired target sites are oriented toward or away from each other, respectively (**Fig 2A**). Upon exonuclease challenge, stretches of gBlock DNA contained between two bound Cas9 enzymes were protected from degradation during a 2-hour incubation, while gBlock stretches not bound on both sides by Cas9 were completely degraded (**Fig 2B**). DNA was protected between two Cas9 enzymes regardless of PAM orientation. However, PAM-in orientation resulted in the highest estimated concentration of the protected segment of DNA following exonuclease challenge (mean PAM-in 225pg/uL, mean PAM-out 106.5pg/uL) and so was selected as the preferable orientation for target enrichment. As expected, in the absence of Cas9, nearly all gBlock DNA is degraded by the three exonucleases (**Fig 2B**).

## Yield and coverage

We targeted 5 loci using the CaBagE method; guide RNAs were selected with "PAM-in" orientation and are listed in **S1 Table**. As a proof of concept, we targeted loci in healthy donor DNA, including a highly variable hexanucleotide repeat in *C9orf72*, and four cancer-related genes with guide RNAs previously validated for PCR-free targeted sequencing (*GSTP1, KRT19, GPX1, SLC12A4*) [16]. We multiplexed up to four loci per reaction and sequenced on a single flow cell. Target enrichment and sequencing for each locus was run in duplicate and runs targeted one or four loci, respectively, on a single flow cell (**Table 1**). Multiplexing multiple loci on a single flow cell did not significantly impact coverage across each individual locus, though coverage did vary from run-to-run.

Sequence reads were aligned using MiniMap2 [32] and on-target reads were visualized with IGV [33]. On-target reads were considered as any reads that overlap the target region by at least 1bp and were counted using samtools [34]. Reads that overlap the target by greater than 90% were considered spanning reads and were counted using the bedtools "coverage" utility [35]. When sequencing across the repeat region of C9orf72 (~4Kb) in a healthy donor, over 90% of on-target reads spanned the locus, terminating at the Cas9 cleavage sites on either side. Further, both DNA strands were equally represented in the alignment data (**Fig 3**). For the

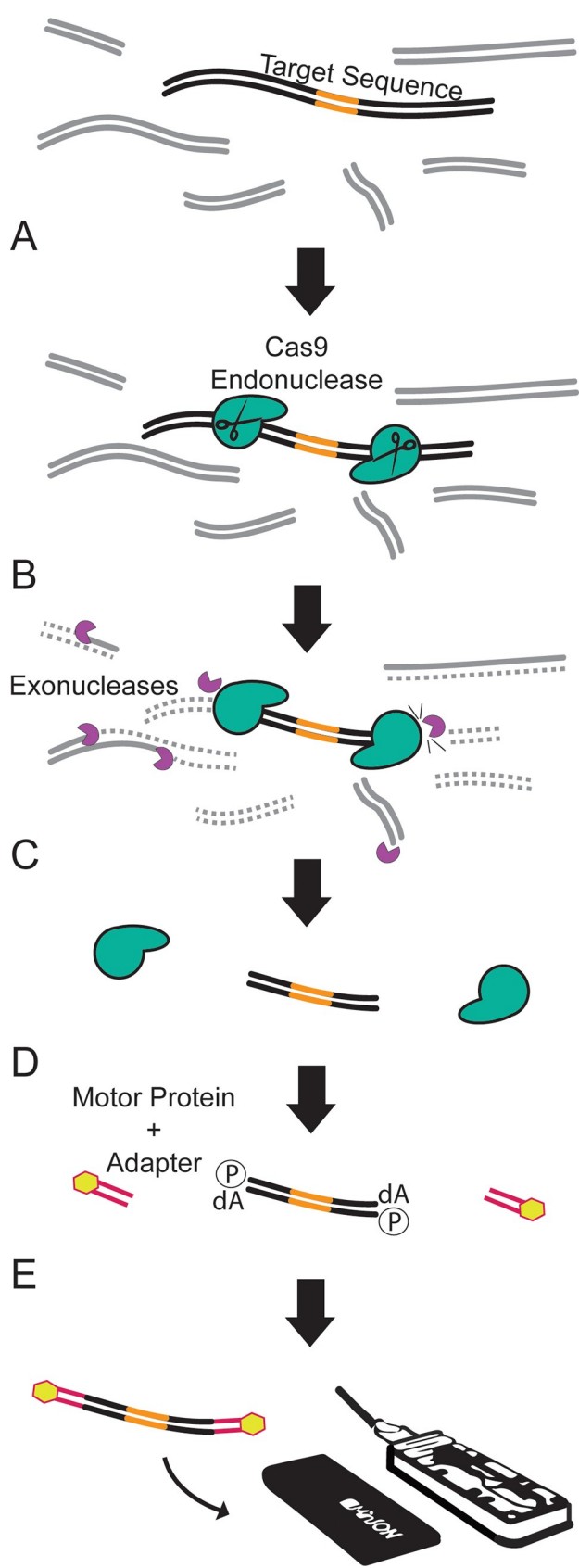

**Fig 1. Schematic of Cas9 background elimination strategy.** A) Cas9 is bound to either side of target sequence. B) Off-target DNA is digested with a combination of exonucleases. C) Heat is used to dissociate that Cas9 and inactivate the exonucleases. D) On-target fragment is available for A-tailing and sequence adapter ligation. E) Target fragments are sequenced on the MinION for 48 hours.

largest target, *SLC12A4* (~24Kb), >65% of on-target reads spanned the locus (**Table 1**). The vast majority of off-target reads were <1,000bp in length. We found that selecting for larger fragments after adapter ligation using the ONT Long Fragment Buffer, which selects for fragments longer than 3kb, resulted in fewer reads overall and fewer on-target reads despite target fragments being larger than 3kb. For example, two independent runs using the same initial DNA sample with Short Fragment Buffer and Long Fragment Buffer generated 2,707,912 reads with 71 on-target and 99,191 reads with 14 on-target, respectively. As expected, the Long Fragment Buffer resulted in an enrichment of longer reads and also higher proportion of reads with map quality ≥60 (**S1 Fig**). However, due to the difference in the number of on-target reads, all CaBagE runs utilize the Short Fragment Buffer. Off-target reads were typically short (median length = 559bp, **Fig 4A**) and randomly distributed throughout the genome, suggesting that they arose primarily by incomplete exonuclease digestion rather than off-target guide RNA binding. To determine whether off-target reads were enriched for other genomic features that might be preferentially protected from exonuclease digestion, we tested for a statistical enrichment for overlaps with G-quadruplex annotations (permutation test, p = 0.97) [36, 37]; further, the GC content distribution of off-target reads centered at 39.5%, reflecting the genome average (**S2 Fig**). Ten genomic regions showed pile-ups with >50X coverage, and these sites were annotated as having long chains of simple tandem repeats; therefore, the pile-ups were likely the result of mapping errors. The total number of reads generated from CaBagE targeted sequencing ranged from ~800,000 to 2.7 million. When restricting to reads with map quality ≥60, ~40% of off-target reads are removed (**Fig 4B**).

To determine how target enrichment with CaBagE compares to nCATs in our hands, side-by-side sequencing runs targeting four loci were conducted. Using identical DNA input

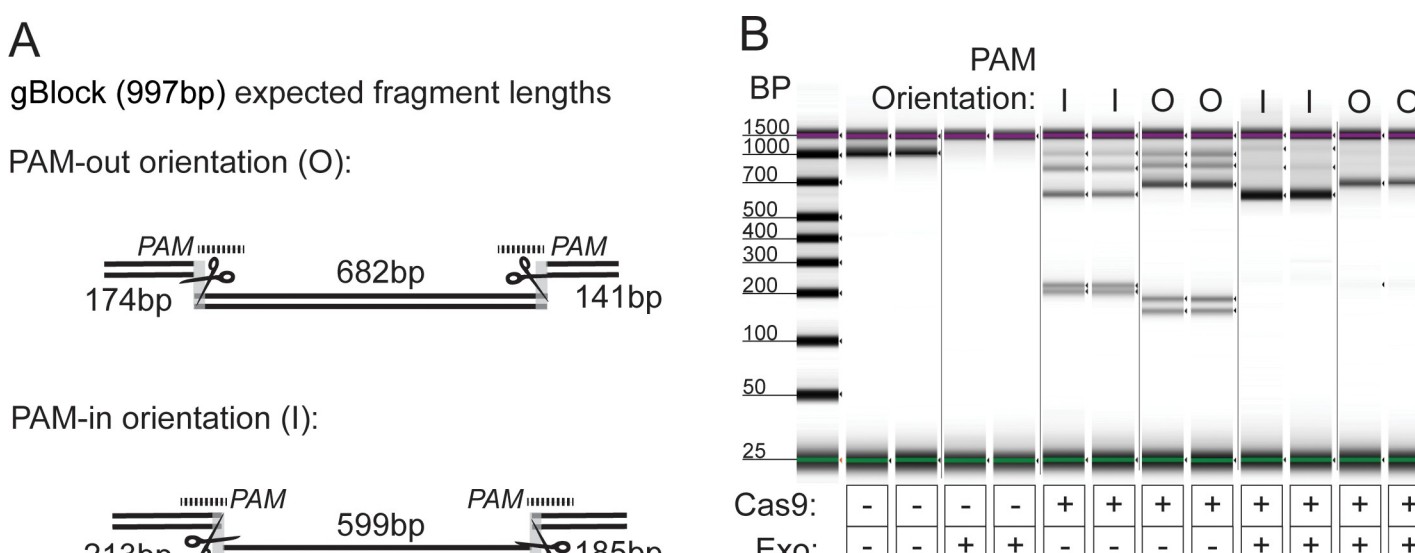

**Fig 2.** A. gBlock assay design for Cas9 challenge with exonuclease. gBlock contained two pairs of gRNA target sites, one with PAM-out orientation and one with PAM-in orientation. Upon Cas9 binding (depicted by scissors), each set of target sites generate 3 unique fragment lengths. The gRNAs are represented as dotted lines. B. Capillary electrophoresis results from exonuclease challenge experiment with Cas9. 15nM gBlock DNA was incubated with 40nM ribonucleoprotien complex, followed by digestion with a combination of exonucleases for 2 hours. When Cas9 is used without exonucleases, the gBlock is cut to produce expected fragment lengths. Upon challenge with exonuclease, only the fragments flanked on both sides by Cas9 remain in the sample. (l = in; O = out).

**Table 1. Results from individual CaBagE runs in DNA from healthy donors.**

| Run ID | Total Reads[a] | Target(s) per flowcell | Target Length (bp) | On-Target Read Depth | Total Spanning Reads[b] |
|--------|------------|------------------------|--------------------|----------------------|-------------------------|
| L1R1 | 536,943 | C9orf72 | 4,044 | 416 | 404 |
| L1R2 | 485,412 | C9orf72 | 4,044 | 179 | 168 |
| L4R1 | 845,510 | GSTP1 | 17,819 | 91 | 61 |
| | | KRT19 | 18,189 | 162 | 98 |
| | | GPX1 | 13,644 | 190 | 136 |
| | | SLC12A4 | 24,389 | 116 | 77 |
| L4R2 | 681,142 | GSTP1 | 17,819 | 39 | 25 |
| | | KRT19 | 18,189 | 61 | 36 |
| | | GPX1 | 13,644 | 54 | 39 |
| | | SLC12A4 | 24,389 | 63 | 41 |

[a]MapQ = 60

[b]Reads that span ≥ 90% of the target locus

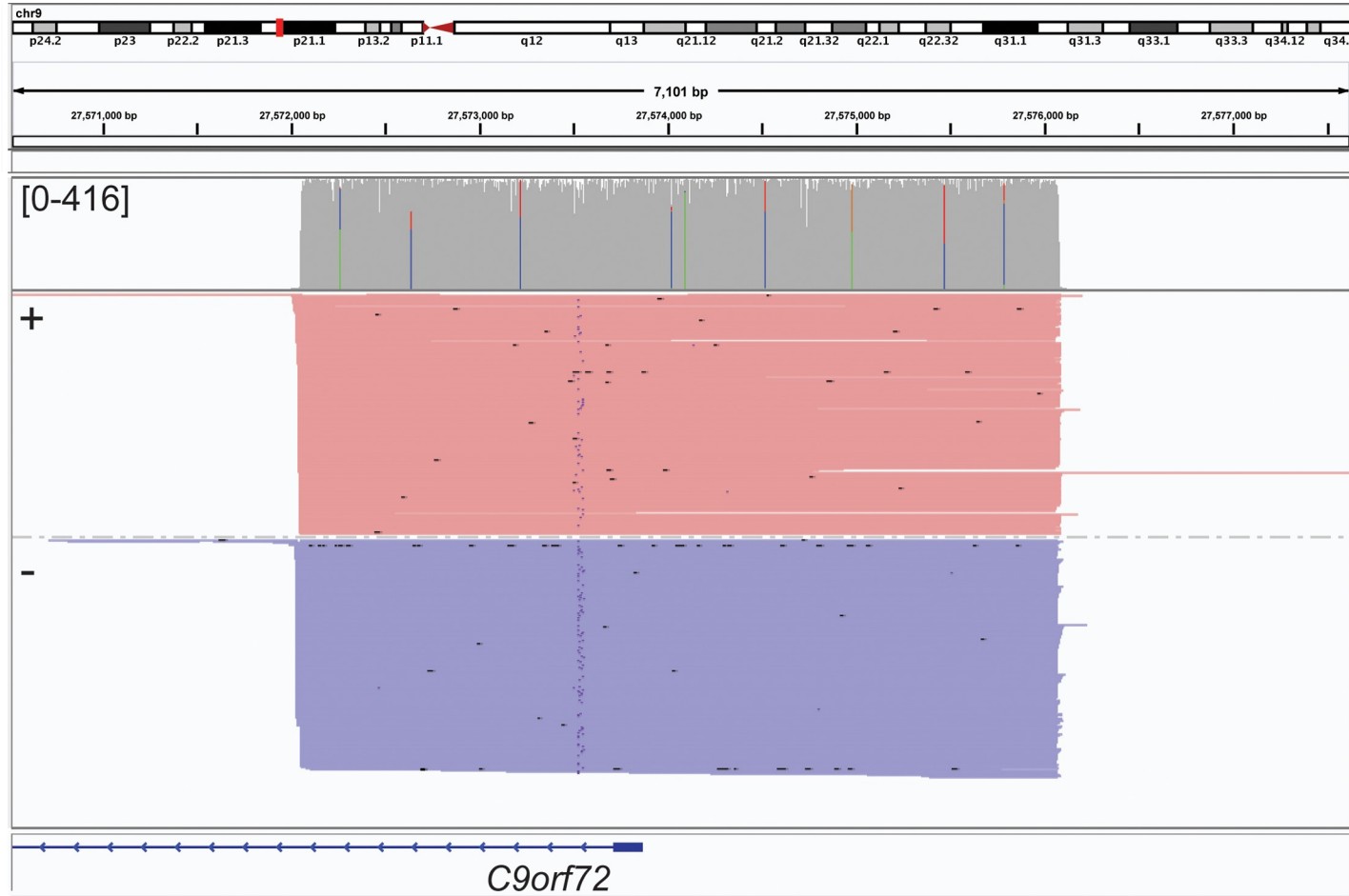

**Fig 3. On-target reads (416X coverage) produced using the CaBagE target sequence enrichment strategy to capture the *C9orf72* repeat-expansion locus in a healthy individual.** IGV screenshot shows aligned reads sorted by strand (plus, red; minus, blue).

A

B

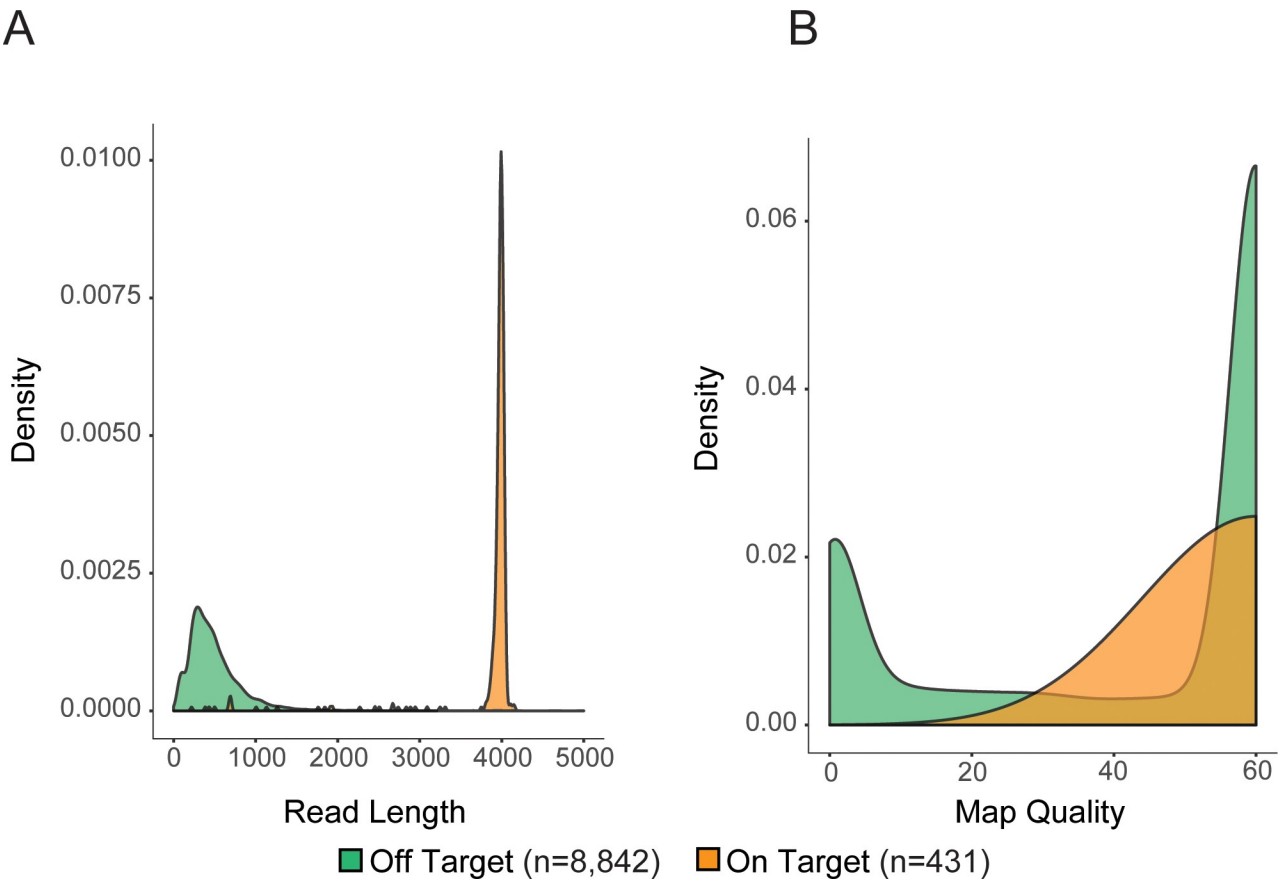

**Fig 4. Characteristics of a random sample of 1% of primary alignments from off-target reads and all on-target reads from a CaBagE run enriching for a 4,044bp target in a healthy individual.** A) Kernel density plot of read lengths in off- and on-target reads B) Kernel density plot of map quality scores in off- and on-target reads.

samples, concentrations, and sequencing parameters on flow cells that performed similarly during Platform QC (i.e. similar number of active pores available) the on-target read depth at the target locus achieved with nCATs was 2.6 to 10.7-fold higher than that of CaBagE (**S2 Table**). While the CaBagE off-target sequencing rate resulting from incomplete exonuclease digestion likely contributed to its relatively lower on-target yield, coverage across the targets produced by CaBagE were sufficiently high (≥30X) for locus characterization.

## CaBagE target enrichment produces reads that span a pathogenic repeat expansion in known carriers

To test the ability of our target enrichment strategy to sequence through disease-specific tandem repeat alleles in affected individuals, we applied CaBagE to two de-identified DNA samples with known *C9orf72* repeat expansions from the National Institute of Neurological Disorders and Stroke (NINDS) repository at the Coriell Institute. Repeat copy numbers for these individuals were previously estimated using gene specific repeat-primed PCR (RP-PCR) and gel electrophoresis [38]. The upper limit of detection for repeat copy number estimation using RP-PCR is ~950 copies and genotypes above 950 copies are denoted as EXP, for expanded [38]. The PCR-based copy-number estimates for the two samples' expanded alleles are 704 and EXP, respectively, where the EXP allele was beyond the upper limit of detection

**Table 2. Results from CaBagE runs in known carriers of the *C9orf72* repeat expansion.**

| Coriell ID | RP-PCR CN Estimate | Total Reads[a] | On-Target Read Depth | Total Spanning Reads | Reads spanning expanded repeat | CaBagE CN Estimate |
|---|---|---|---|---|---|---|
| ND11386 | 8/704 | 1,490,712 | 115 | 98 | 21 | 9/749/1,893 |
| ND13803 | 2/EXP | 852,155 | 71 | 66 | 7 | 2/808/1,538 |

[a]MapQ = 60

*RP-PCR repeat-primed PCR and agarose gel electrophoresis derived genotypes from Bram *et al* [38], CN copy number

with PCR-based methods. Targeted sequencing of the *C9orf72* repeat expansion using the CaBagE method in these individuals resulted in high (>60X) depth of coverage at the target locus (**Table 2**). A bias for the minus strand was observed in both NINDS ALS samples (**Fig 5**). Strand bias has been previously observed when sequencing across repeats with ONT [39, 40] and can be correlated with repeat length, however we observed no apparent relationship between strand and repeat size. The G-rich and C-rich repeats of sense and antisense ssDNA at this locus form different secondary structures, which may migrate through the sequencing pores at different rates [41].

Spanning reads were defined as reads that aligned to both the 5 prime and 3 prime flanking sequence around the repeat, as well as the full repeat sequence itself. Per-read hexanucleotide repeat copy number was estimated by counting the number of bases between the position in the read that aligned immediately upstream of the repeat and immediately downstream, divided by six, the repeat motif length. Allele-specific repeat copy numbers were estimated from subgroup means derived from a Gaussian mixture model where the number of clusters was determined a priori by visually counting distinct peaks from a read-length histogram. In both samples, the read-length histograms showed 3 populations of spanning read lengths (**Fig 5**) and triallelic repeat copy number estimates are listed in **Table 2**.

In sample ND11386, the majority of the expanded reads supported a copy number estimate 749 (**Fig 5A**) and for ND13803, the majority of expanded reads supported a copy number 1,538 (**Fig 5C**), consistent with the estimates derived from RP-PCR. In both samples, the largest alleles detected were absent from the RP-PCR results, as they are larger than the detectable limit of the assay. Further, both samples showed a strong bias to sequencing the shortest allele, representing 79% and 91% of the spanning reads, respectively. This is likely an artifact of the technology sequencing shorter fragments more efficiently, as has been previously observed [19, 42, 43] and the fact that longer (e.g. expanded) fragments are more likely to be damaged between the flanking Cas9 binding sites, which would result in failure of enrichment. The presence of the three alleles in each sample were confirmed by repeated library preparation and sequencing of the same samples (**S3 Fig**). The appearance of the third alleles in these samples could be artifacts of cell line transformation from which the DNA was derived. Multiple populations of allele lengths have been previously observed in cell lines and was observed in ND11836 via Southern blot during validation of a PCR-based assay [44].

## Discussion

We developed a method to enrich long-read sequence data for specific target loci that is fast, efficient, and amenable to the multiplexing of multiple target loci. By relying on the binding kinetics of the Cas9 enzyme to its RNA-guided target, CaBagE can flexibly enrich for targets so long as most fragments in the input DNA are intact between Cas9 binding sites. Therefore, to pursue very large targets (>~30Kb) will likely require ultra-high molecular weight DNA, which must be obtained with specialized DNA extraction methods such as agarose plugs or ultra-high molecular weight DNA extraction kits.

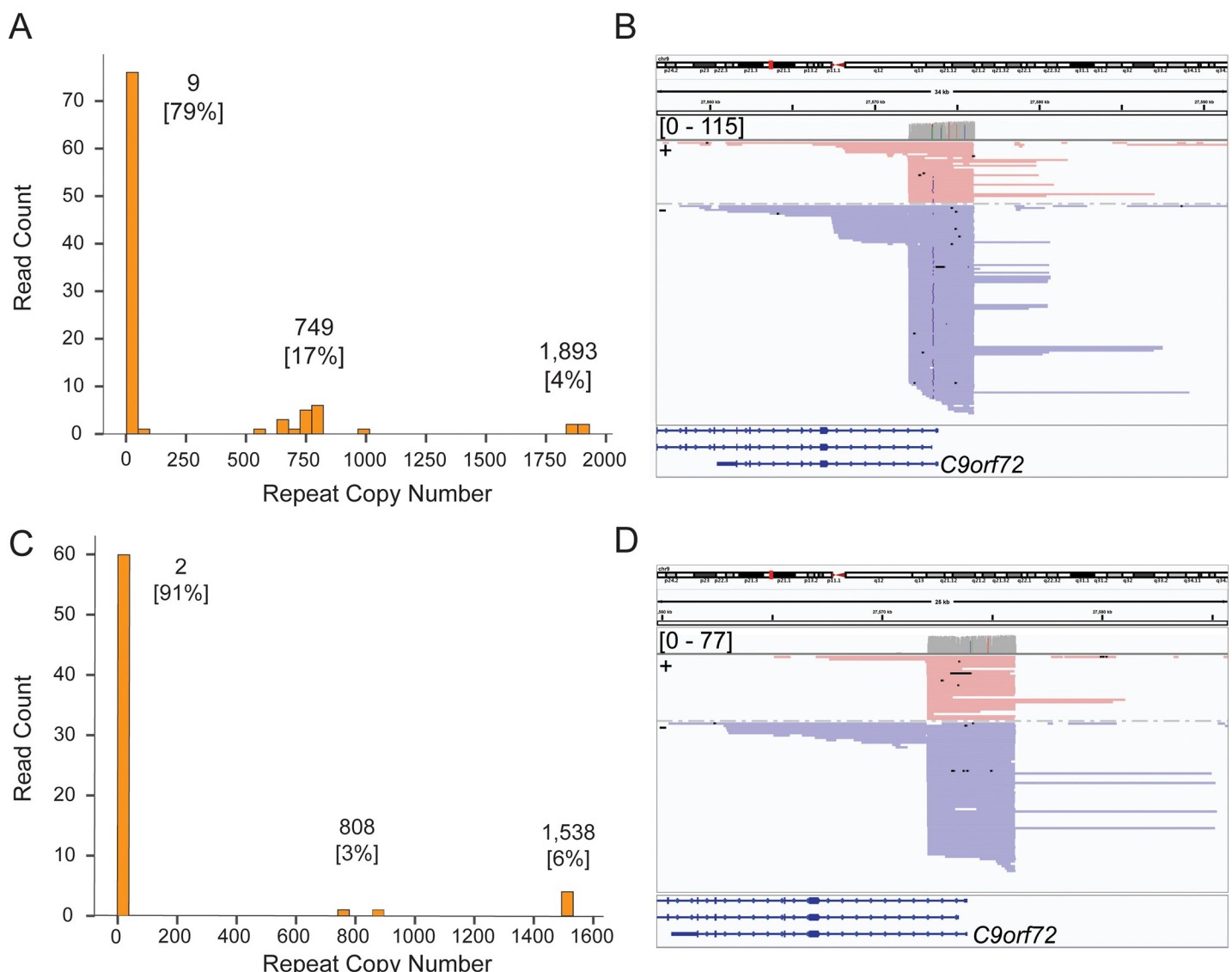

**Fig 5. Targeted sequencing across repeat expansion at *C9orf72* in two ALS cases.** A) Histogram of repeat copy number distribution and copy number estimates derived from a Gaussian mixture model for ND11836 (copy number, [percent of on-target reads]). B) IGV screen shot showing expanded reads across the hexanucleotide repeat for subject ND11836. C) Histogram of repeat copy number distribution and copy number estimates derived from a Gaussian mixture model for ND13803 (copy number, [percent of on-target reads]). D) IGV screen shot showing expanded reads across the hexanucleotide repeat for subject ND13803.

CaBagE performs similarly in terms of prep time and input requirements, but with a lower yield than a popular competing method, nCATS [16]. Specifically, CaBagE costs approximately $9.40 more per run than nCATs and requires two additional hours of hands-off incubation time. The reduction in yield that we observe is most likely driven by the inefficiency of exonuclease digestion relative to dephosphorylation, which could be improved with further optimization of the protocol. There is also an increased sensitivity of CaBagE to fragmentation between Cas9 binding sites, where any break in DNA or failure of binding by either of the guides will result in degradation of the target molecule. This sensitivity to breakage increases with increased target size, which is reflected in **Table 1**, where the overall yield and proportion of reads that span the target is lower in larger targets. However, unlike the nCATS and ReadFish methods for amplification-free targeted sequencing, the enrichment achieved from CaBagE

occurs at the DNA-level, where the ratio of on- to off-target DNA physically increases in the sample prior to sequencing. The Negative Enrichment strategy shares this feature of CaBagE, however, CaBagE utilizes a larger DNA input, different exonucleases and shorter digestion time, as well as modifications to the library preparation, which lead to significantly higher on-target coverage after sequencing on the MinION (3-32-fold higher). Physically enriching DNA for a specific target without modifying native DNA using CaBagE may therefore prove useful for applications beyond long-read DNA sequencing where isolating specific DNA sequence is required. Furthermore, while a Southern blot is the current gold standard for diagnosis of several repeat expansion disorders, it requires high sensitivity and low background caused by non-specific binding of the probe. The physical removal of off-target DNA by CaBagE might prove useful in background reduction for the Southern Blot and increase specificity for other size selection applications. Physical enrichment of target DNA in a sample may also aid in PCR-free cloning. For example, transformation-associated recombination (TAR) cloning is a method where efficiency has already been shown to increase with the introduction of double-strand breaks around the target of interest (~2% vs. ~30% gene-positive colonies) [45]. This efficiency may be further increased with the simple addition of the CaBagE background elimination step.

Despite high on-target coverage, CaBagE sequences off-target fragments at a high rate owing to both incomplete exonuclease digestion and the lack of a selection step for long fragments. However, since an average CaBagE run yields ~1 Gb of sequence, which is well under the >8 Gb typical throughput for the MinION R9.4.1 using the ligation kit, we expect this high off-target rate isn't detracting from our on-target depth.

We demonstrated CaBagE's ability to capture pathogenic repeat-expansion alleles in two ALS patients. We discovered 3 distinct read-length populations in each sample, potentially representing significant mosaicism. This observation is not uncommon in studies of repeat expansions where genotyping assays are performed on cell line-derived DNA [44, 46]. Determining whether these 3 alleles were present in the blood of these patients or arose as an artifact of cell culture or sequencing would require both blood and LCL-derived DNA from the same individual, which is not available for the NINDS ALS Collection.

We note that several challenges remain in utilizing targeted long-read sequencing in the identification of repeat expansions. First, longer repeat expansions have greater instability, and growing and shrinking of repeat length is common and variable cell-to-cell and tissue-to-tissue in patients with the *C9orf72* repeat expansion and other repeat expansion diseases [47, 48]. The observation of mosaic lengths of short tandem repeats in ours and previous studies poses an interesting challenge for estimating repeat-length genotypes and further calls into question whether creating a consensus sequence for the repeat is biologically meaningful. However, estimating a distribution of repeat lengths within an individual may be of clinical relevance, where a greater spread may indicate instability, which in turn may be correlated with pathogenesis. Second, sequencing across the repeat expansion using CaBagE resulted in a strong bias in the sequencing data toward shorter alleles. Therefore, in addition to needing high depth of coverage to detect the expansion, this length bias also complicates the ability to accurately quantify relative clonal contributions in cases where somatic mosaicism is present. Carefully extracted, high molecular weight DNA may not have as pronounced a bias, as longer fragments won't be depleted in those samples. Overcoming this bias would be required for future studies of mosaicism. Accurate base calling also remains a challenge using ONT technologies, particularly in repeats with high GC content. We note that some reads representing the expanded alleles failed base calling using Guppy and were retrieved from the "fastq_fail" folder generated by the MinKNOW software. As the performance of Guppy continues to improve, methods that have been developed to detect tandem repeat in long-read sequencing data will also improve. For example, STRique [19] and TRiCoLoR [49], which detect repeat expansions from aligned

reads, have already outpaced Nanosatellite, a repeat detection algorithm designed to circumvent issues with base calling by detecting repeats from raw signal data [42]. Strand biases are also exacerbated across repeats sequenced with long-read technologies [39] and should be considered during repeat sequence characterization.

CaBagE's amplification-free targeted sequencing can be used to effectively sequence across multiple, large loci on a single MinION flow cell. The method is not limited to the MinION, but should be adaptable to any long-read sequencing technology. Future work to improve the method will include increasing the efficiency of the exonuclease digestion and possibly adapting the method to be used for tiling across much larger targets with catalytically inactive dCas9. CaBagE is a target enrichment strategy that does not simply enrich sequencing data for specific loci, but enriches the DNA sample itself without amplification, thus potentially providing utility beyond long-read sequencing. As methods for DNA preparation, sequencing, and downstream data processing continue to improve, targeted sequencing methods like CaBagE will become indispensable in large-scale, cost-effective studies of complex structural variation.

## Methods

### Samples

*A* 997bp gBlock was designed to contain four gRNA target sites (**S1 Table**). Deidentified healthy donor DNA was obtained from Promega (Human Genomic DNA: Female, G152A). DNA from ALS cases (ND11836 and ND13803) were extracted from EBV transformed LCLs by from the National Institute of Neurological Disorders and Stroke (NINDS) repository at the Coriell Institute. DNA was pre-treated with FFPE Repair Mix from NEB (M6630S) according to manufacturer's *Protocol for use with Other User-supplied Library Construction Reagents* to repair nicks that could result in undesired target degradation by exonucleases.

### Guide RNA design

Guide RNAs (sgRNA, **S1 Table**) were selected to flank up and downstream of the target locus. A combination of online tools including CHOPCHOP, E-CRISP, and IDT [50–52] were used to design sgRNAs with high *in silico* predicted on-target efficiency and minimal off-target effects. For target loci, pairs of sgRNAs were designed such that they maintained a "PAM-in" orientation to the target sequence. Preassembled gRNA comprised of crRNA and tracrRNA (IDT, Alt-R® CRISPR-Cas9 sgRNA, 2 nmol) sequences were purchased from IDT and resuspended in IDTE at a 10μM concentration.

### Cas9 digestion

The molar ratio of Cas9:gRNA:DNA target was ~10:10:1. The ribonucleoprotein complex was formed by combining 150nM Cas9 enzyme with 150nM of each guide in 1X CutSmart buffer (NEB) and the 23.5μL reaction was incubated at 25°C for 10 minutes. A 40uL reaction containing the RNP complex, ~15nM (3ug) human genomic DNA or 30ng of gBlock in 1x Cutsmart buffer (NEB B7204) was incubated at 37°C for 15 minutes.

### Exonuclease digestion

Immediately following Cas9 digestion, 260 total units of exonucleases (Exo I ([40U] NEB M0293), Exo III ([200U] NEB M0206), Lambda ([20U] NEB M0262)) diluted in 1X CutSmart buffer to 10μL were added to the reaction for a final reaction volume 50uL and incubated at 37°C for two hours, followed by heat inactivation at 80°C for 20 minutes.

## A-tailing

1μL of 10mm dATP (Zymo Research, D1005) and 1μL Taq DNA Polymerase (M0267S) were added to reaction mix and incubated at 72˚C for 5 minutes.

## Adapter ligation

An adapter ligation mix was prepared from the LSK-109 Ligation Sequencing Kit by combining 25μL Ligation Buffer, 5μL Quick T4 Ligase (NEB E6057), 5μL Adapter Mix, and 13μL nuclease-free water. The mixture was added to the previous reaction for a total volume of 100uL and incubated for 10 minutes on a hula mixer at room temperature. A clean-up step was then performed using 0.3X AmpureXP magnetic beads (Beckman Coulter A63881) and washed twice with 200μL of Short Fragment Buffer (ONT SQK-LSK109). The final library was eluted in 16.6μL of Elution Buffer and 15.8μL retained.

## Nanopore sequencing

Each sample was sequenced on a MinION flow cell (R9.4.1). Flow cells with >800 active pores following Platform QC were primed according to the adapted protocol from Gilpatrick et al [24] with 800μL of Flush Buffer followed by a second priming with priming mix (70μL Sequencing Buffer + 70μL nuclease-free water + 70μL Flush Buffer). The final library is then immediately loaded onto the flow cell in a mixture with 26μL Sequencing Buffer, 9.5μL Loading Beads, and 0.5μL Sequencing Tether from the LSK-109 Ligation Kit. Sequencing was performed for 48 hours using default settings with the MinKNOW software (v.19.05.0) and live base calling was conducted using the high accuracy flip-flop algorithm.

## Sequence data alignment and QC

All sequencing reads were aligned to the human reference GRCh38 using minimap2 software with parameters (-Yax map-ont) appropriate for ONT and to prevent hard clipping of supplementary alignments [53]. Reads were considered on-target if they overlapped the target locus by at least 1 bp. Spanning reads aligned to the >90% of the target between Cas9 cleavage sites. Off-target reads with mapQ = 60 were counted using samtools v.1.9. On-target depth of coverage was also measured with samtools and visualized in IGV. GC content of all off-target reads was calculated using samtools and awk and compared to a random sample of 1,000,000 intervals in the GRCH38 reference using Bedtools "nuc" (v2.28.0). All off-target reads were also tested for enrichment with secondary structure annotations, namely G-quadruplexes, using poverlap [37], which permutes a null distribution of overlapping genomic regions.

## Repeat copy number estimation in ALS samples

On-target reads at the C9orf72 locus were identified using samtools by identifying reads that overlap the target locus by at least one base pair [34]. For large expansions, a single read would often be soft-clipped within the repeat with sequence up- and downstream represented as multiple alignments in the resulting BAM file.

On-target reads were realigned to the upstream and downstream sequences flanking the repeat expansion using the Striped Smith-Waterman algorithm to determine whether the read completely spanned the repeat (scikit-bio v.0.2.3 [54], Python v.2.7). Repeat-spanning reads were defined as reads that aligned both 10bp upstream and 10bp downstream of the repeat after realignment.

To determine repeat copy length, the base pair position representing the end of the alignment to the upstream flank was subtracted from the start position of the alignment to the downstream

flank within each repeat-spanning read. The repeat length was divided by 6 (the repeat unit length) to estimate repeat copy number. Reads that failed base calling were also aligned with Striped Smith-Waterman to ensure that we weren't missing on-target reads where the repeat interfered with base calling. Repeat length distributions were then visualized on a histogram to determine the number of expected clusters of allele-lengths, which were then fed into a Gaussian Mixture Model (scikit-learn 0.22.1 [55]) to determine allele-specific repeat copy number estimates.

### Accession numbers

All sequencing data from healthy donors are available on the Sequence Read Archive under accessions PRJNA687491. Data from two ALS cases is available through dbGaP with accession phs002368.v1.p1.

Data, analysis code and a detailed wet laboratory protocol used to generate the results for this manuscript are available at https://github.com/adw222/CaBagE-manuscript.

### Supporting information

**S1 Fig. Read length and quality using short and long fragment buffer.** Characteristics of a random sample of 9000 reads produced from a CaBagE run enriching for a 4,044bp target. The experiment was conducted in tandem using the same sample DNA with the ONT Long Fragment Buffer (LFB) during adapter ligation or with the Short Fragment Buffer (SFB). A) Kernel density plot of read lengths in LFB and SFB reads. B) Kernel Density plot of map quality scores in LFB and SFB reads.
(TIF)

**S2 Fig. GC content of off-target reads.** GC content distribution of all off-target reads from a single CaBagE run (n = 890,627) compared to a random 1,000,000 intervals from GRCh38 with length equal to the mean off-target read length of the CaBagE run.
(TIF)

**S3 Fig. Replicates of C9orf72 repeat copy number estimates in expansion carriers.** Histograms of repeat copy number distributions for replicated target enrichment and sequencing across C9orf72 repeat expansions in two individuals with ALS. Results confirm presence of >2 alleles in both individuals.
(TIF)

**S1 Table. Guide RNA sequences.**
(XLSX)

**S2 Table. Comparison of coverage across targets for CaBagE and nCATs.**
(XLSX)

**S1 Raw images.**
(PDF)

### Acknowledgments

We would like to thank the following individuals for their expertise and effort in all aspects of this study: Nels Elde for sharing laboratory space and resources, Joe Brown for hardware support, and Simone Longo and Harriet Dashnow for editing the manuscript.

### Author Contributions

**Conceptualization:** Amelia D. Wallace, Katherine E. Varley, Aaron R. Quinlan.

**Data curation:** Amelia D. Wallace.

**Formal analysis:** Amelia D. Wallace, Brooke L. Gates, Jeff Greenland.

**Funding acquisition:** Amelia D. Wallace, Aaron R. Quinlan.

**Investigation:** Amelia D. Wallace, Thomas A. Sasani, Jordan Swanier, Brooke L. Gates, Aaron R. Quinlan.

**Methodology:** Amelia D. Wallace, Jordan Swanier, Jeff Greenland, Aaron R. Quinlan.

**Project administration:** Amelia D. Wallace.

**Resources:** Katherine E. Varley, Aaron R. Quinlan.

**Software:** Brent S. Pedersen, Aaron R. Quinlan.

**Supervision:** Katherine E. Varley, Aaron R. Quinlan.

**Validation:** Amelia D. Wallace.

**Visualization:** Amelia D. Wallace.

**Writing – original draft:** Amelia D. Wallace, Aaron R. Quinlan.

**Writing – review & editing:** Amelia D. Wallace, Thomas A. Sasani, Jordan Swanier, Brooke L. Gates, Jeff Greenland, Brent S. Pedersen, Katherine E. Varley, Aaron R. Quinlan.

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
