## [Decision Letter · Decision Letter 0]

2 Nov 2020

PONE-D-20-31316

CaBagE: a Cas9-based Background Elimination strategy for targeted, long-read DNA sequencing

PLOS ONE

Dear Dr. Quinlan,

Thank you for submitting your manuscript to PLOS ONE. After careful consideration, we feel that it has merit but does not fully meet PLOS ONE’s publication criteria as it currently stands. Therefore, we invite you to submit a revised version of the manuscript that addresses the points raised during the review process.

The expert reviewers were generally positive about your paper, but indicated that the approach was not entirely innovative. Please compare your methods with previously published similar methods of Cas9 sequence enrichment. Please present median read depth in the abstract. Please pay attention to the remaining useful recommendations of the reviewers that will strengthen your submission.

We look forward to receiving your revised manuscript.

Kind regards,

Alfred S Lewin, Ph.D.

Academic Editor

PLOS ONE

Journal Requirements:

Reviewers' comments:

Reviewer's Responses to Questions

**Comments to the Author**

1. Is the manuscript technically sound, and do the data support the conclusions?

Reviewer #1: Yes

Reviewer #2: Yes

2. Has the statistical analysis been performed appropriately and rigorously? 

Reviewer #1: Yes

Reviewer #2: Yes

3. Have the authors made all data underlying the findings in their manuscript fully available?

Reviewer #1: No

Reviewer #2: No

4. Is the manuscript presented in an intelligible fashion and written in standard English?

Reviewer #1: Yes

Reviewer #2: Yes

5. Review Comments to the Author

Reviewer #1: This paper presents the CaBagE strategy. The method exploits the stable binding of Cas9 complexes to a target DNA combined with exonucleases digestion to deplete off-target DNA and physically enrich for the region of interest prior to sequencing. The authors evaluate their enrichment strategy on several targets using sequencing runs performed on MinION. Overall the paper reads well and the evaluation of the method uses reasonable experiments.

I am not a wet-lab biologist, so I've decided not to go in-depth on the wet-lab part. I will rather focus on the broader use of CaBagE. Comments below are roughly given in order of decreasing importance. I suppose these can all be addressed without performing additional experiments.

Comment #1. (General comment) The enrichment strategy described in this paper (that is, Cas9 complexes to flank target specific sequences and exonuclease digestion to deplete off-target DNA), has already been described elsewhere (https://doi.org/10.1371/journal.pone.0215441) and termed Negative Enrichment. I'm aware that the lack of novelty is not a major issue with PLOS ONE. However, the author should at least cite the original method in the Introduction section and highlight similarities/differences with their CaBagE strategy, if any (otherwise, the novelty claim should be toned-down as well).

Comment #2. (Table 1 and Abstract, line 63). The authors show that a 416X on-target read depth can be obtained in a single sequencing run after CaBagE enrichment. However, looking at the on-target read depth values in Table 1, it is clear that this event is pretty rare (the median and the mean both equal to 179, less than half of the maximum read depth value). Median (or mean) read depth should be stated clearly in the Abstract in order not to create inflated expectations.

Comment #3. (Results, line 210, and Figure 3). The authors state that any reads overlapping the target region by at least 1 bp were considered for the estimation of the region-specific depth of coverage. I'm not convinced that counting all the reads intersecting the region of interest (even those having just 1 bp falling in the target region) make sense, especially if a genetic variant (that is, a complex structural variant or a tandem repeat expansion) lies in the middle of the target region. Can the authors elaborate on the number of reads overlapping a larger portion of the target region (for instance, 50% of the region of interest)? It would be nice to have a column specifying such a value for each sequencing run/region of interest in Table 1. I guess the authors have these data handy. In addition, as from Figure 3, it seems to me that most of the reads are actually spanning more than 50% of the target region and I guess that these additional data won't deteriorate the performances of CaBagE.

Comment #4. (Results, line 209). Which minimap2 parameters do the authors used to align sequenced reads? In principle, the standard presets for Nanopore Sequencing data (-x map-ont) are not expected to give the best alignment results. Indeed, long expansions relative to the reference may not map through the repeat region if this penalty increases with length (as from the standard --gap-extend and --lj-min-ratio values): in this case, the alignment can get clipped somewhere within the region, leaving only one side of the read mapped.

Comment #5. (Introduction, lines 127-132). The authors cite the ReadUntil method for real-time sequences identification in nanopore sequencing runs. The method has recently been renamed to ReadFish (https://github.com/LooseLab/readfish) and the name should be changed accordingly in the text. Furthermore, it should be stated clearly that only off-target reads are, in principle, rejected (or on-target reads retained).

Comment #6. (Discussion, line 324). The authors state that NanoSatellite may be an effective alternative for repeated sequences characterization as the performance of Guppy improves. This is not completely true. As the authors point out, NanoSatellite was originally built to solve issues with guppy/albacore basecalling by operating on raw signals. As basecalling performances improve, this is nowadays not that effective (see also README.md at https://github.com/arnederoeck/NanoSatellite) and NanoSatellite is not currently mantained (see also https://github.com/arnederoeck/NanoSatellite/issues/14). Other tools for tandem repeat profiling which will benefit from improvements of Guppy basecalling performances exist and, among these, TRiCoLOR (https://doi.org/10.1093/gigascience/giaa101) seem in high-quality.

Reviewer #2: Reviewer's comments attached in a doc file. Overall a nice manuscript that will help researchers interested in applying in vitro CRISPR-Cas9 enrichment to isolate long loci containing complex structural variants

6. PLOS authors have the option to publish the peer review history of their article (what does this mean?). If published, this will include your full peer review and any attached files.

Reviewer #1: No

Reviewer #2: No

---

## [Author Response · Author response to Decision Letter 0]

12 Jan 2021

We have attached a file with our response to reviewers.

---

## [Decision Letter · Decision Letter 1]

20 Jan 2021

CaBagE: a Cas9-based Background Elimination strategy for targeted, long-read DNA sequencing

PONE-D-20-31316R1

Dear Dr. Quinlan,

We’re pleased to inform you that your manuscript has been judged scientifically suitable for publication and will be formally accepted for publication once it meets all outstanding technical requirements.

Kind regards,

Alfred S Lewin, Ph.D.

Section Editor

PLOS ONE

Additional Editor Comments (optional):

Reviewers' comments:

Reviewer's Responses to Questions

**Comments to the Author**

1. If the authors have adequately addressed your comments raised in a previous round of review and you feel that this manuscript is now acceptable for publication, you may indicate that here to bypass the “Comments to the Author” section, enter your conflict of interest statement in the “Confidential to Editor” section, and submit your "Accept" recommendation.

Reviewer #1: All comments have been addressed

Reviewer #2: All comments have been addressed

2. Is the manuscript technically sound, and do the data support the conclusions?

Reviewer #1: Yes

Reviewer #2: Yes

3. Has the statistical analysis been performed appropriately and rigorously? 

Reviewer #1: Yes

Reviewer #2: Yes

4. Have the authors made all data underlying the findings in their manuscript fully available?

Reviewer #1: No

Reviewer #2: Yes

5. Is the manuscript presented in an intelligible fashion and written in standard English?

Reviewer #1: Yes

Reviewer #2: Yes

6. Review Comments to the Author

Reviewer #1: I thank the authors for addressing all of my previous comments. Thanks also to the other reviewer, that helped a lot improving the wet-lab part. I have no further comments.

Reviewer #2: Thank you for taking the time in considering all my comments. I see the authors didn't re-do the tandem repeat profiling with TRICOLOR software as suggested by the other reviewer, however I understand that there is always new software coming to update the ones used. It is up to the other reviewer if the response they gave regarding this, it is sufficient or not. I consider that authors made a great effort in addressing all our comments. I think that now the paper has strengthened and it is ready for publication.

7. PLOS authors have the option to publish the peer review history of their article (what does this mean?). If published, this will include your full peer review and any attached files.

Reviewer #1: No

Reviewer #2: **Yes: **Elena Lopez-Girona

---

## [Editor Report · Acceptance letter]

30 Mar 2021

PONE-D-20-31316R1 

CaBagE: a Cas9-based Background Elimination strategy for targeted, long-read DNA sequencing 

Dear Dr. Quinlan:

I'm pleased to inform you that your manuscript has been deemed suitable for publication in PLOS ONE. Congratulations! Your manuscript is now with our production department. 

Kind regards, 

on behalf of

Dr. Alfred S Lewin 

Section Editor

PLOS ONE